# Cytocompatibility of Graphene Monolayer and Its Impact on Focal Cell Adhesion, Mitochondrial Morphology and Activity in BALB/3T3 Fibroblasts

**DOI:** 10.3390/ma14030643

**Published:** 2021-01-30

**Authors:** Iwona Lasocka, Lidia Szulc-Dąbrowska, Michał Skibniewski, Ewa Skibniewska, Karolina Gregorczyk-Zboroch, Iwona Pasternak, Marie Hubalek Kalbacova

**Affiliations:** 1Department of Biology of Animal Environment, Institute of Animal Science, Warsaw University of Life Sciences, 02-786 Warsaw, Poland; iwona_lasocka@sggw.edu.pl; 2Department of Preclinical Sciences, Institute of Veterinary Medicine, Warsaw University of Life Sciences, 02-786 Warsaw, Poland; lidia_szulc-dabrowska@sggw.edu.pl (L.S.-D.); karolina_gregorczyk-zboroch@sggw.edu.pl (K.G.-Z.); 3Department of Morphological Sciences, Institute of Veterinary Medicine, Warsaw University of Life Sciences, 02-776 Warsaw, Poland; 4Faculty of Physics, Warsaw University of Technology, 00-662 Warsaw, Poland; iwona.pasternak@pw.edu.pl; 5Faculty of Medicine Pilsen, Charles University Prague, Cz-121 Prague, Czech Republic; marie.kalbacova@lf1.cuni.cz

**Keywords:** graphene, fibroblast, cytocompatibility, focal contact, mitochondria

## Abstract

This study investigates the effect of graphene scaffold on morphology, viability, cytoskeleton, focal contacts, mitochondrial network morphology and activity in BALB/3T3 fibroblasts and provides new data on biocompatibility of the “graphene-family nanomaterials”. We used graphene monolayer applied onto glass cover slide by electrochemical delamination method and regular glass cover slide, as a reference. The morphology of fibroblasts growing on graphene was unaltered, and the cell viability was 95% compared to control cells on non-coated glass slide. There was no significant difference in the cell size (spreading) between both groups studied. Graphene platform significantly increased BALB/3T3 cell mitochondrial activity (WST-8 test) compared to glass substrate. To demonstrate the variability in focal contacts pattern, the effect of graphene on vinculin was examined, which revealed a significant increase in focal contact size comparing to control-glass slide. There was no disruption in mitochondrial network morphology, which was branched and well connected in relation to the control group. Evaluation of the JC-1 red/green fluorescence intensity ratio revealed similar levels of mitochondrial membrane potential in cells growing on graphene-coated and uncoated slides. These results indicate that graphene monolayer scaffold is cytocompatible with connective tissue cells examined and could be beneficial for tissue engineering therapy.

## 1. Introduction

Medical materials have to be tested in vitro to choose the most suitable of them for further studies including in vivo tests with animal models and finally clinical trials with human patients. In vitro studies provide a rapid answer whether the material selected for testing—in our experiment graphene substrate –will be non-toxic and have positive effects on cell physiology. Cytotoxicity assessment is the first step in evaluating biocompatibility of the given material. ISO (Organization of Standardization) 10993 standards are well established and give guidance on testing the biocompatibility of materials; especially, ISO 10993 Part 5 describes usual techniques to evaluate cytotoxicity of materials [1]. Therefore, these methods and techniques have been adopted by researchers to evaluate the biocompatibility of nanomaterials [2]. This investigation will be helpful in the evaluation of tissue engineering technologies using graphene as a raw material. There are many in vitro cytotoxicity tests, which detect the toxic potential of the material studied [3]. Results of these tests could be expressed as cell viability, rate of proliferation or rate of metabolism. The most frequently used cells are of rodent or human origin. It should be noted that the more various cell lines are used in biocompatibility tests the wider the answer about medical potential of material will be. Park et al. [4] suggested that in vitro cytotoxicity test of biomaterials should be compared to other cytotoxicity testing methods with the use of multiple cell lines. Fibroblasts are among the most common cell types in the body and are widely used in cytotoxicity studies [5]. There are three well established methods of contacting studied material with cell: direct contact, indirect contact, and elution of test material. In direct contact method the cells are seeded on top of the material or material is placed on top of the cells. This method was used in our experiment to evaluate the influence of graphene monolayer substrate on BALB/3T3 murine cells. Our previous study [6] aimed to assess the biocompatibility of the graphene substrate as a scaffold for L929 cells. The two abovementioned cell lines are the most often used in cytotoxicity tests [7,8]. Moreover, it is recommended to perform the in vitro tests on the same type of cells which will be in direct contact with the medical materials in vivo. We suppose that graphene scaffold for fibroblasts or mesenchymal stem cells could be used in the future to aid reconstruction of damaged tissue caused by mechanical injuries, burns or as a result of chronic diseases.

Graphene has unique properties that make it a suitable candidate for several biomedical applications. High electrical conductivity, thermal stability, mechanical strength and antibacterial activity of graphene promote research into its use in the biomedical field [9]. Graphene and its derivatives are used in research in drug delivery, tissue engineering and anti-cancer therapy [10]. Because of its properties, the intensity of research on “graphene family” biomaterials and their application in regenerative medicine is rapidly increasing.

Fibroblasts play a crucial role in wound healing and are one of the major cells within the tissue that contact to the implanted device. Therefore, we used BALB/3T3 cells, derived from mouse embryos, for assessment of pristine graphene biocompatibility. Cytocompatibility of graphene monolayer was analyzed based on the assessment of BABL/3T3 cell viability, size and morphology. Additionally, because formation of focal adhesions allows cells throughout integrins to adsorb on the substrate surface and mediate signaling between cell and environment, we also assessed the graphene effect on stabilization of focal cell contacts in fibroblasts. Focal adhesions (FA) are mature opposite to nascent focal complexes, and are generally larger than 1 μm^2^ [11]. FA can also be classified as dot and dash adhesions, which means associated with cell motility or stability, respectively. Depending on the length, FA can be defined as nascent focal complexes (˂1 μm long), focal adhesions (1–5 μm long) and supermature or fibrillar adhesions (>5 μm long) [12]. Young and Higgs [13] revealed that FA consists of focal adhesion unit (FAU) 300 nm wide and the FA width is defined by the number of FAU but the FA length is variable and highly dynamic. Focal contacts were analyzed to evaluate the influence of graphene substrate. BALB/3T3 cell morphology (spreading, attachment, cell lysis) was examined under a light optical microscope, viability by trypan blue exclusion test, focal contacts by immunofluorescence staining and mitochondrial activity by colorimetric assay (WST-8) using a spectrophotometer. Moreover, mitochondrial network morphology using fluorescent dyes and membrane potential by flow cytometry was examined.

Lastly, in the present work we evaluated the impact of graphene monolayer on mitochondria morphology and physiology in BALB/3T3 cells. Mitochondria form a complex network in the cell that is subject to continuous fusion and fragmentation processes [14]. Mitochondria are involved in cellular processes, i.e., ATP synthesis, apoptosis or buffering of calcium ions in the cell [15,16]. It has been shown that graphene oxide in the form of nanoplates can pass into the cytosol, where it can interact with mitochondria, and affect their morphology, function and membrane potential [10,17]. However, it should be remembered that the interaction between graphene and the cell membrane is closely related to the physicochemical properties of graphene, including size, shape and chemical forms [6].

## 2. Materials and Methods

### 2.1. Graphene Monolayer Scaffold

In this work, we used graphene supplied from Graphenea company, San Sebastian, Spain. Graphene was transferred onto rounded glass cover slides (1 cm of diameter and 0.17 mm of thickness) from copper substrate. The high-speed (1 mm/s) electrochemical delamination method was used in the transfer process. Cover slides coated with graphene were sterilized by UV (30 min on the both sides) before experiment and then placed into the well of the 24-well plate. The characterization of the properties of graphene transferred onto glass substrate was performed at room temperature by Raman spectroscopy using a Renishaw in Via Raman microscope system with a 532 nm Nd:YAG laser as an excitation source. The Raman spectrum presented in Figure 1a contains two prominent G and 2D peaks and a negligible low, disorder-related D peak near 1340 cm^−1^. The observed symmetric Lorentzian line shape of the 2D peak and its low Full Width at Half Maximum (FWHM about 29 cm^−1^) are features confirming the presence of predominantly single layer graphene [18,19,20,21]. Moreover, the micro-Raman map of the intensity ratio of the 2D to the G peaks (I_2D_/I_G_) in the area of 40 × 40 μm^2^ taken with a step of 1 μm is shown in Figure 1b. The average value (see inset of Figure 1b) close to 5 indicates existence of predominantly a monolayer graphene structure on glass substrate.

Micro-Raman map of the I_2D_/I_G_ together with optical image (Figure 2) of the graphene layer depict that transferred graphene is a continuous layer.

### 2.2. Cytotoxicity Assessment of Graphene Scaffold

Cytotoxicity was assessed by direct contact method as per ISO-10993-5 guideline [1]. The BALB/3T3 cells were seeded on the glass cover slide with graphene (experimental group) and without graphene (control group). Three independent experiments were done in triplicate. The degree of toxicity of the graphene monolayer was assessed on the basis of changes in cell morphology, viability (Trypan Blue test) and mitochondrial network morphology (Mito Green and Mito Red), mitochondrial activity (Water Soluble Tetrazolium-WST-8 test) and mitochondrial membrane potential (JC-1).

### 2.3. Cell Culture and Experimental Design

BALB/3T3 clone A31 cells (ATCC CCL-163) were purchased from the American Tissue Culture Collection (ATCC, Manassas, VA, USA) and cultured in high glucose Dulbecco’s modified Eagle’s medium (DMEM; HyClone, Logan, UT, USA) supplemented with 10% iron-enriched bovine calf serum (BCS, HyClone). Additionally, culture medium was supplemented with antibiotics: 100 U/mL penicillin and 100 μg/mL streptomycin (Sigma-Aldrich, Saint Louis, MO, USA), and antimycotic: 0.25 μg/mL amphotericin B (Sigma-Aldrich). Cells were maintained at 37 °C in a humidified atmosphere of 5% CO_2_ in air. Confluent monolayers were detached by incubating with 0.25% trypsin-EDTA solution (Sigma-Aldrich). For experiments, BALB/3T3 cells (3 × 10^4^/well) were seeded onto sterile microscopic glass coverslips (control) and graphene-coated glass coverslips placed in 24-well plates (Falcon, Los Angeles, CA, USA). After 24 h, cells were harvested for further experiments.

### 2.4. Cell Morphology Evaluation

Morphology of BALB/3T3 fibroblasts was observed at 1, 3, 6, 12 and 24 h of cell seeding on control or graphene-coated microscopic slides. Images were captured with an inverted microscope (Olympus IX71) equipped with Color View III cooled CCD camera and Cell^F software (Soft Imaging System) (Olympus, Tokyo, Japan).

### 2.5. Trypan Blue Cell Viability Analysis

BALB/3T3 cells grown for 24 h on glass coverslips (control) or graphene-coated microscopic slides (experimental) were detached with trypsin-EDTA solution (HyClone, Logan, UT, USA). After centrifugation, collected cells were resuspended in DMEM and mixed 1:1 with a 0.4% Trypan Blue solution (Sigma-Aldrich), and the mixture was transferred to a Neubauer chamber. Blue (dead) and unstained (live) cells were then counted, and these counts were used to calculate the relative proportion of the dead cells. The results were expressed as a percentage of viable cells.

### 2.6. Water Soluble Tetrazolium (WST) Assay

Cell Counting Kit-8 (Sigma-Aldrich) is a colorimetric, cytotoxicity assay for measuring mitochondrial activity.

Standard curve was prepared by plating 100 µl of 2-fold serially diluted suspension of BALB/3T3 cells into wells of the sterile flat-bottom 96-well plate (Falcon, Bedford, VA, USA). The ranges of the standard curve sensitivity for BALB/3T3 fibroblasts were approximately 1.5 × 10^4^–1.17 × 10^2^ cells/mL. Negative control containing media only (no cells) was used for measurement of the background. Next, 10 µl of Cell Counting Kit-8 reagent was added to each well of the standard curve and incubate for 2 h at 37 °C in a humidified atmosphere of 5% CO_2_ in the air. The absorbance of triplicate serial dilutions in 96-well plate was determined at 450 nm using an Epoch Microplate Spectrophotometer (BioTek, Winooski, VT, USA). All experimental analysis were performed in parallel with the standard curve.

BALB/3T3 cells grown for 24 h on glass coverslips (control) or graphene-coated glass coverslips (experimental) were detached with trypsin-EDTA solution and resuspended in appropriate culture medium. Cells in the total volume of 100 μL were plated in triplicate into the wells of 96-well plate. 10 µL WST-8 reagent was added to each well (three control and three experimental for both cell lines), and the plates were incubated at 37 °C in a humidified CO_2_ incubator for 2 h. The absorbance was determined at 450 nm using a spectrophotometer.

### 2.7. Cell Area (Spreading) Calculation

BALB/3T3 cell area after 12 and 24 h of incubation on glass and graphene substrate was calculated using the ImageJ software (NIH, Bethesda, MD, USA).

### 2.8. Immunofluorescent Staining and Morphometric Analysis of Focal Contacts

To detect F-actin and vinculin, BALB/3T3 cells were fixed with 4% PFA in PBS for 20 min. Next, the cells were permeabilized with 0.5% Triton X (Sigma-Aldrich) in PBS (15 min) and blocked with 3% bovine serum albumin (BSA, Sigma-Aldrich) in 0.1% Triton X-100-PBS solution (30 min) to prevent nonspecific binding. For F-actin detection, the cells were incubated with 0.5 µg/mL phalloidin-FITC for 20 min in the dark. For immunostaining of focal contacts, anti-vinculin antibody (Sigma) was used. The cells were incubated with primary Abs anti-vinculin for 1 h and next, secondary anti-mouse Abs conjugated with rhodamine Red-X were used (60 min in the dark). DNA was stained with 1 µg/mL Hoechst 33342 for 5 min in the dark. Slides were mounted in ProLong Gold Antifade Reagent (ThermoFisher Scientific, Waltham, MA, USA).

For quantitative analysis of focal contacts, fluorescent images were collected using Color View III cooled CCD camera mounted on a fluorescence microscope (Olympus BX60). Focal contact number per cell and the average focal contact size were evaluated using Cell^F (Olympus, Tokyo, Japan) and ImageJ software (NIH Image, version 1.50i, Bethesda, MD, USA). Focal contacts were manually outlined, and area (size) was calculated automatically. For focal contact number per cell at least 50 cells of each experiment were counted, and for focal contacts size, at least 10 areas of focal contacts per cell were analyzed.

### 2.9. Mitochondrial Network Morphology Evaluation and Determination of Mitochondrial Membrane Potential

BALB 3T3 cells were seeded on glass coverslips coated or uncoated with graphene (control) in a 24-well plate and mitochondria were labeled with 300 nM Mito Red or Mito Tracker Green for 30 min in the dark at 37 °C with 5% CO_2_ in a humidified incubator. Next, the staining solution was replaced with fresh prewarmed media, and mitochondrial network was observed under Olympus BX60 microscope.

Mitochondrial membrane potential was measured with the LSR Fortessa flow cytometer (Becton Dickinson Biosciences) by using the dual-emission potential-sensitive probe JC-1 as a fluorescent dye. JC-1 exhibits potential-dependent accumulation in mitochondria where it starts forming J aggregates (red fluorescence with emission at ~590 nm); upon depolarization, it remains as monomer (showing green fluorescence with emission at ~529 nm). Mitochondrial depolarization is indicated by a decrease in the red/green fluorescence intensity ratio.

BALB 3T3 cells grown on graphene-coated or uncoated glass slides in 24-well culture plates were incubated with culture medium supplemented with (6 μL do 1 mL) µM JC-1 dye, for 60 min in the dark at 37 °C with 5% CO2 in a humidified atmosphere. Next, the cells were collected, resuspended in cold (4 °C) PBS and analyzed by flow cytometry (λex = 488 nm). Data were presented as the ratio of red to green fluorescence of JC-1 dye. For positive control of decreased mitochondrial membrane potential, the cells were treated with both JC-1 and 3% hydrogen peroxide.

## 3. Results

### 3.1. Morphology

The assessment of the in vitro cytotoxicity of the substance or scaffold is often a qualitative analysis based on the morphological examination of cell damage and growth after direct contact with the material [1,3,22]. In general, qualitative assessing occurs by evaluating morphological aspects of cells microscopically, such as shape and size, adhesion/detachment, cell viability/lysis or cell membrane damage/integrity.

The cytotoxicity testing using BALB/3T3 mouse cells over 24 h reveals the attachment and growth of fibroblasts on the graphene scaffold in a manner that is comparable to that on the control substrate. During the first 3 h of incubation, cell adhesion to the substrate (graphene and glass) and gradual flattening of cells were observed. Starting from 6 h of incubation onset of changes in their morphology was noticed (Figure 3). BALB/3T3 fibroblasts began to change shape to more spindle-like and extend pseudopods from various parts of its spherical body. At 12 h most of the cells were spindle-shaped with numerous pseudopods (Figure 3). Microscopic examination revealed single nuclei with well-formed nucleoli. At 24 h of incubation in most cell pseudopods gradually merge into the thin peripheral plate called lamella.

The BALB/3T3 cells were visualized to grow directly on the graphene scaffold without damage or changes in morphology or detachment and presented a similar morphology (Figure 3) to the control cell culture.

### 3.2. Size of BALB/3T3 Cells

BALB/3T3 cells exhibit contact inhibition. This phenomenon is known as contact paralysis and means that cell do not move over the surface of another cell and turn away when its active edge touch the surface of neighboring cell. That is why it is so important to culture BALB/3T3 cells at the right density to fully express their morphology and achieve proper size. For 24 h experiment BALB/3T3 cells were planted at a density 3 × 10^4^ cells per cm^2^, as described in the Materials and Methods. After 12 h of incubation average size of cells cultured on graphene and control substrate was 1638.6 ± 709.5 and 1732.8 ± 711.2 µm^2^, respectively (Figure 4). Over the next 12 h of incubation, area of cells growing on both substrates increased nearly by half (Figure 4). There was no significant difference in the size of BALB/3T3 cells cultured on glass and graphene. In our previous study using L929, we have also not shown such changes [6].

### 3.3. Focal Cell Adhesion

Focal contacts display an ellipsoidal shape and were elongated in the same direction of corresponding actin stress fibers (Figure 5).

We can distinguish large focal adhesions tended to be elongated and small, round, probably—nascent focal complexes (Figure 4). Higher focal contact area was noted on graphene substrate. Moreover the shape resembled a combination of several ellipses in contrary to control group were individual ellipses predominated. There was also significant difference in number of focal contacts of cells growing on glass and graphene. On graphene substrate, there were less focal contacts but with a larger single focal contact size compared to control-glass substrate.

### 3.4. Viability and Mitochondrial Activity

The Trypan Blue Exclusion test is used to examine cell viability, where dead cells are stained because the non-toxic dye cannot permeate intact cell membranes. Viabilities of cell growing on control substrate (glass) and graphene scaffold were 95% and 94% (Figure 6), respectively.

Viabilities of greater than or equal to 95% are considered as excellent. The WST-8 assay results reveal that the mitochondrial activity of BALB/3T3 cells significantly (*p* ˂ 0.01) increased with graphene as a scaffold for cells, as shown in Figure 6. Mitochondrial activity raised from 100 ± 3.6% in control cells grown on glass substrate to 122 ± 5.8% in cells seeded on graphene scaffold.

In vitro studies of graphene substrate in direct contact method did not show any toxic effects on BALB/3T3 cells. It can be concluded that the use of graphene substrate enables the adhesion and enhance the mitochondrial activity of BALB/3T3 cells.

### 3.5. Mitochondrial Network Morphology

The morphology of mitochondrial network brings important information about the cell health and its function. Therefore, to visualize mitochondrial morphology we used two dyes—Mito Tracker Green FM (in living cells) and Mito Red (in fixed cells). There were no changes in the mitochondrial network morphology and distribution in BALB/3T3 cells grown on graphene. Images of cells growing on both media—glass and graphene—were characterized by an even distribution of the mitochondrial network and well connected (Figure 7).

The mitochondrial network was neither too fragmented nor too elongated and does not show swollen and irregular structures or giant spherical mitochondria. Mitochondria were oriented parallel to the long axis of BALB/3T3 cells. Mitochondria of cells growing on control substrate and graphene can be described as networked and rod-like with different phenotypes: straight rods, twisted rods, branched rods and loops (“donut”).

### 3.6. Mitochondria Membrane Potential

Flow cytometry evaluation of mitochondrial membrane potential using JC-1 dye clearly showed that graphene did not cause decrease of this parameter in BALB/3T3 cells. The percentage of cells with low (green color) and high (red color) membrane potential in the control and graphene groups was comparable. More cells with reduced potential (blue color) were shown in the control group versus the group with graphene coated slides (Figure 8).

Nearly 100% of H_2_O_2_-treated cells showed a clear and significant decrease in the mitochondrial membrane potential (blue color). Additionally, cells exposed to H_2_O_2_ showed hyperfragmentation of the mitochondrial network, as determined by fluorescence microscopy (Figure 9).

There were no longer string shaped mitochondria-like in control group and graphene substrate, instead punctate and swollen mitochondria occurred. Meanwhile, between cells cultured on graphene-coated and uncoated slides there were no visible changes in the level of green or red fluorescence of JC-1.

## 4. Discussion

Graphene has potential to be used in medical fields and composite enhancement, amongst other uses. Biosafety of nanomaterials has caused increased attention from scientists who are investigating their effects on the cells, animals and environment [23,24,25]. Comparative studies on graphene cytotoxicity help to efficiently apply these materials in medical fields. That is why the main goal of this study was to determine the cytotoxicity of graphene by in vitro tests on murine BALB/3T3 fibroblast. The research provides additional data on the suitability of graphene monolayer for being used as a scaffold for cells in regenerative medicine. Previously, we checked biocompatibility of pristine graphene with L929 fibroblast cells [6]. The reason for choosing another cell type was to see if the effect of cytotoxicity was cell dependent.

The success of tissue engineering scaffolds highly depends on their interaction with cells. First of all, scaffold cannot be toxic to the tissue in which it is going to be implanted. Furthermore, it is desirable for the scaffold to stimulate physiologic changes leading to, especially in fibroblasts, accelerated cell migration into the wound and increase in proliferation and viability. The exposure of BALB/3T3 cells to graphene scaffold induces a high increase of average cell mitochondrial activity, implying that the graphene scaffold is not cytotoxic and stimulates cells proliferation. Proliferating cells reduce their internal environment more than non-proliferating ones do. The increase in mitochondrial activity by 1/5 compared to control indicates that graphene scaffold stimulates proliferation of cells attached to this nano-material. This is in agreement with results obtained by Gentile et al. [26], who observed that nano-topography of the material on which cells move affects the physiology of said cells, modifying viability and proliferation. Biggs et al. [27], on the basis of literature review, concluded that nano-topography plays an essential role in the creation of focal adhesion and subsequent changes in cellular functions. Gentile et al. [26] noticed that moderately rough (average surface roughness between 10 and 40 nm) surfaces of electrochemically etched silicon substrates increased the proliferation rate of NIH/3T3 fibroblasts from 36 to 60 h after seeding. Mitochondrial activity of BALB/3T3 introduced to graphene raised significantly 24 h after seeding. It can be supposed that the mechanism of graphene action is a result of its topography and roughness, which cause increased effective surface energy. Nanoscale topography of substrates and their stiffness became the subject of many studies trying to estimate the possibility of their use as a platform for cell culture [26,28,29,30]. They established that cell adhesion, proliferation and differentiation were influenced by the micro- and nano-surface characteristics of biomaterials. Kim et al. [29] also indicate that graphene oxide may enhance the affinity of cells. In vitro cytotoxicity tests performed in our study reveal excellent cytocompatibility of graphene scaffold to BALB/3T3 cells. Nishida et al. [30] have published results that confirm our data, but they involve other cells—MC3t3-E1 (mouse osteoblastic cells) as well as a different scaffold, using graphene oxide attached to collagen substrate. The authors noted that proliferation of MC3t3-E1 on the scaffold was stimulated in dose-dependent manner by GO application. Moreover, they concluded that the graphene oxide scaffold exhibited good biocompatibility based on implantation of 1 μg/mL graphene oxide scaffold into the subcutaneous tissue of Wistar rats and to extraction sockets of beagle dogs.

Our results presented actin cytoskeleton staining and showed longitudinal filaments along the long axis of the BALB/3T3 cells with colocalization of vinculin on the cell edges. Vinculin is an important FA protein, characteristic for nascent cell-matrix adhesion and mature FA, which bound integrins to cytoplasmic F-actin [31]. Disruption of the vinculin-F-actin interaction affects cell motility, cell stiffness and adhesion [32]. Zhou et al. [31] revealed that vinculin in focal adhesion plaques was significantly decreased in response to soft substrate compared to stiff substrate and cell spreading areas of chondrocytes, osteoblasts, osteoclasts, osteocytes and bone marrow-derived stem cells were also reduced in the soft substrate. Kim and Wirts [33] characterized parameters of FA of mouse embryonic fibroblasts—MEFs and found that FA display an ellipsoidal shape and FA area was approximately 3 μm^2^ for glass substrate and decreased with the decreasing of substrate stiffness, reaching about 1.5 μm^2^ on soft gel.

Average focal adhesion size for different cell lines (CHO, C2H12 and MDCK) growing on flat surfaces was larger than 1 μm^2^; complexes smaller than 1 μm^2^ were identified by authors as nascent focal complexes and were located preferentially in the lamellipodium [11]. Moreover focal adhesion of cells growing on nanopillars was much smaller than for cells growing on a flat surface and was between 0.2 and 0.4 μm^2^. Focal contacts in our experiment were also included in these two morphologies: smaller and larger contacts. Young and Higgs [13] found that focal adhesions initially appear as assemblies of multiple linear units (FAU) of 0.3 μm width, which can split in coordination with FA elongation and that dynamic interactions between FAU control adhesion morphology. The authors have pointed out that, apart from splitting, mature FA can disassemble or forego splitting altogether. Our results indicated larger focal contacts in fibroblasts growing on graphene which could be explained by interactions with FAU. We can assume that focal contact in graphene group consisted of more FAUs than in control group. Mechanism of the FAU connecting on the graphene substrate should be the subject of further studies.

Changes in mitochondrial morphology could be a good indicator of cell condition [34]. Although mitochondrial morphology is variable and can be described as reticulated or fragmented with spheroid-shaped mitochondria or branched structure. The right proportion of certain types of mitochondria is important to deduce their condition and function. It also depends on the cell type and cell cycle. Mitochondrial shapes fall into four “categories”: point, rod-like, branched, and large and round [35]. Mitochondria in BALB/3T3 cells growing on graphene-coated or uncoated glass slides exist as a branched network with no significant morphological dysfunction. Symptoms of lesions or defects in mitochondrial morphology include swelling of mitochondria and giant spherical mitochondria [36]. Mitochondrial morphology also changes as a result of fission and fusion (from fragmented to highly elongated) [37]. Moreover, mitochondrial shape can change many times over time and as a result of stress factors [15,34]. Highly fragmented mitochondria were noted in BALB/3T3 treated with H_2_O_2_ (positive control). This oxidative stress-induced reagent, which caused mitochondrial fission resulted in fragmented mitochondria. This fragmentation was also reflected in highly reduced mitochondrial potential (Figure x). Jaworski et al. [17] found that platelets of graphene have dose-dependent cytotoxicity via depletion of the mitochondrial membrane potential against human cancer cells. Monolayer of graphene—used in this study, in contrast to graphene platelets and H_2_O_2_, has no toxic effect on network and mitochondrial potential.

Cytotoxicity testing of graphene as a biomaterial is mandatory in the assessment of its safety. It should be noted that the biocompatibility of graphene will vary strongly with its type: monolayer graphene, graphene oxide, reduced graphene or few layer graphene [29]. The cytocompatibility of the graphene scaffold with BALB/3T3 cells suggest its possible use in the healing of tissue damage.

## 5. Conclusions

Our study has lead us to conclusion that graphene monolayer has no noticeable cytotoxic effect to BALB/3T3 cell culture. Well spread cytoskeleton with long lamellae protruding out of the cell membranes and high mitochondrial activity indicated healthy, growing cells. Lack of negative impact of graphene on the fibroblasts suggests that this nanomaterial can be used as a scaffold for reconstructive tissue engineering.

## Figures and Tables

**Figure 1 materials-14-00643-f001:**
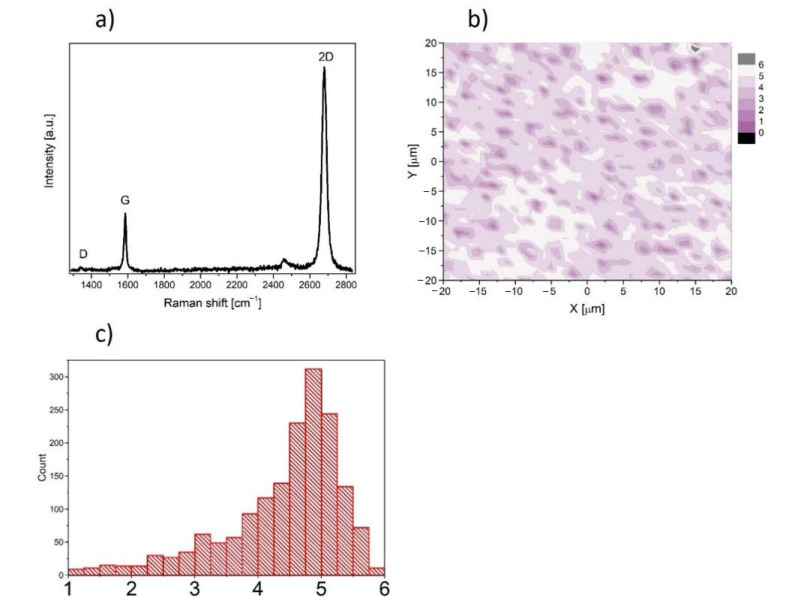
(**a**) Raman spectrum of graphene film on glass substrate. (**b**) Micro-Raman map of intensity ratio of the 2D to the G peaks. (**c**) histogram of the intensity ratio of the 2D to the G peaks.

**Figure 2 materials-14-00643-f002:**
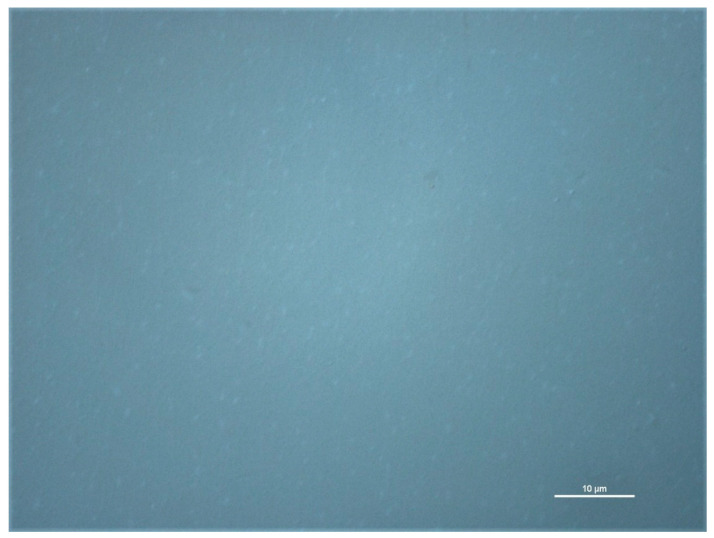
Optical image of graphene layer transferred onto glass substrate. Scale bar is 10 µm.

**Figure 3 materials-14-00643-f003:**
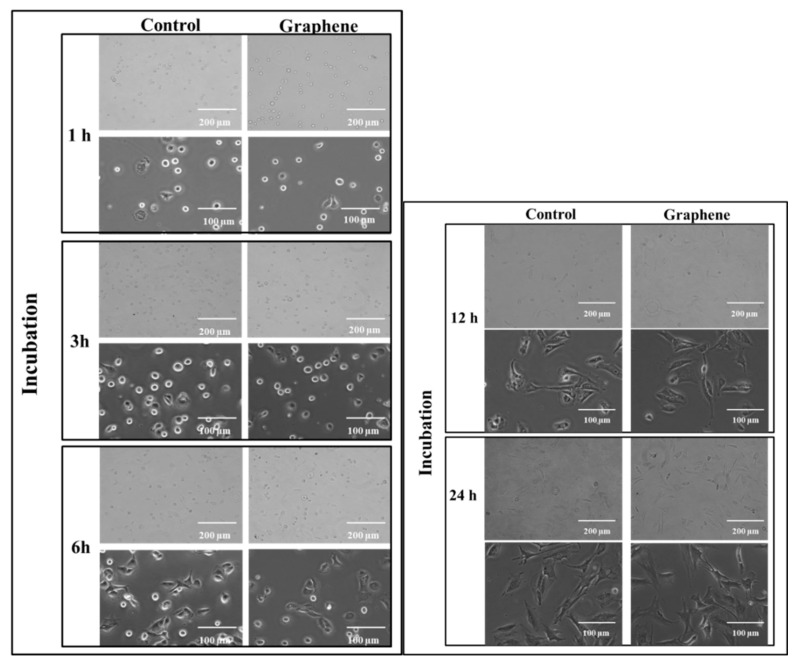
Morphology of BALB 3T3 cells growing on glass and graphene substrate at different time of incubation 1, 3, 6, 12 and 24 h.

**Figure 4 materials-14-00643-f004:**
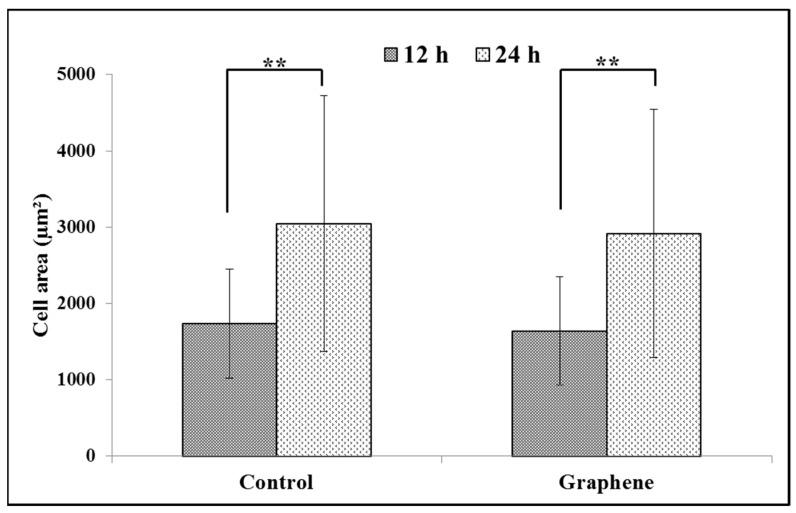
Spreading (cell area) of BALB/3T3 fibroblasts 12 and 24 h after seeding on glass and graphene platform. Data are represented as mean ± SD (standard deviation) of three independent experiments. Approximately 70 cells were evaluated in each experiment. Single cell-area was measured using ImageJ software. **-statistically significant differences.

**Figure 5 materials-14-00643-f005:**
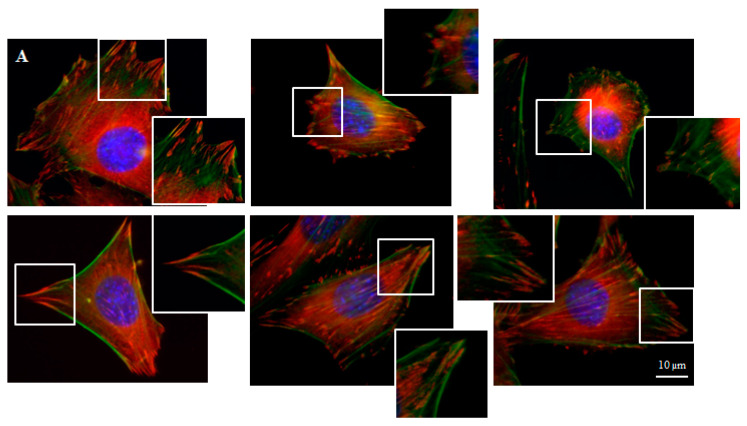
Focal contacts morphology of BALB 3T3 cells and calculation. Vinculin staining of focal contact on control (**A**) and graphene substrate (**B**). Focal contact number and size area of BALB 3T3 cells (**C**,**D**). Representative vinculin staining of attachment points (focal contacts) localized in lammelum area. The arrow indicates the direction of cell movement (**E**). Cells were stained with phalloidin-FITC (F-actin, green fluorescence), anti-vinculin Abs (vinculin, red fluorescence) and Hoechst 33,342 (DNA, blue fluorescence); Magnification in squares presents focal contacts and interaction between vinculin and actin. For focal contact number per cell at least 50 cells of each experiment were counted, and for focal contacts size, at least 10 areas of focal contacts per cell were analyzed. Data from three independent experiments are presented as mean ± SD (standard deviation). **-statistically significant differences.

**Figure 6 materials-14-00643-f006:**
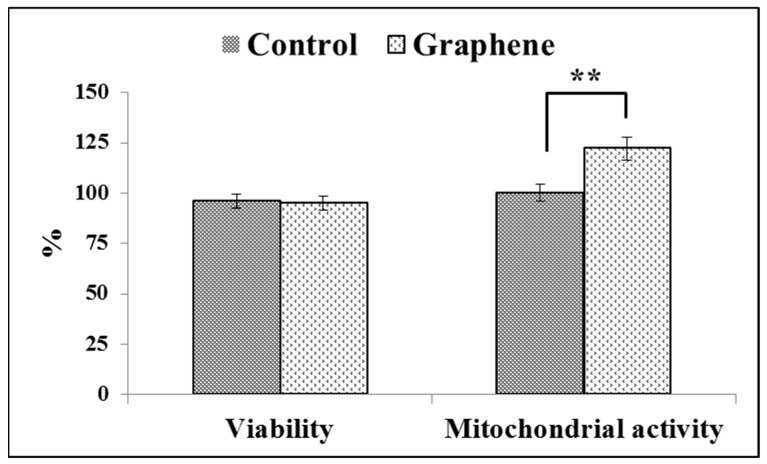
Trypan blue and WST-8 assay results on the cell viability and mitochondrial activity of BALB 3T3 cells on the test material. Data from three independent experiments are presented as mean ± SD (standard deviation). **-statistically significant differences.

**Figure 7 materials-14-00643-f007:**
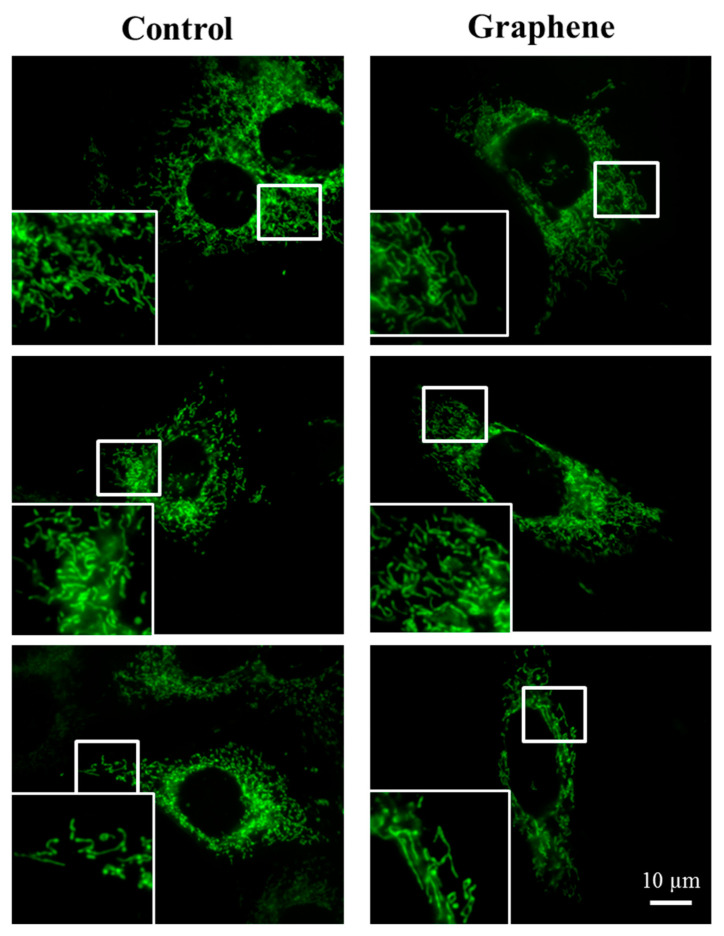
The morphology of mitochondrial network after staining with Mito Tracker Green FM and Mito Red Green fluorescence in living cells and red fluorescence in fixed cells. Magnification in squares presents different phenotypes of mitochondria: straight rods, twisted rods, branched rods and loops.

**Figure 8 materials-14-00643-f008:**
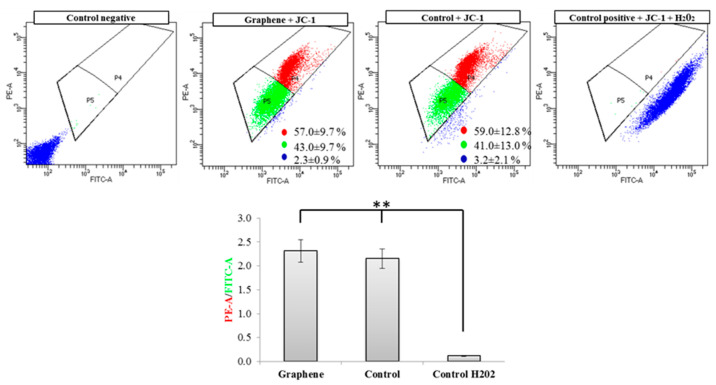
Evaluation of mitochondrial membrane potential using JC-1 dye. Cytograms of JC-1-stained cells; Red populations—cells with high membrane potential. Green populations—cells with low membrane potential. Blue populations—cells with high mitochondrial depolarization. Bar chart of red/green fluorescence intensity ratio of JC-1-stained cells. Data from three independent experiments are presented as mean ± SD (standard deviation) (n = 10,000 cells). ** - statistically significant differences.

**Figure 9 materials-14-00643-f009:**
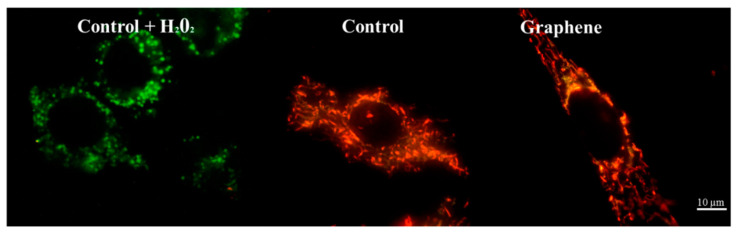
Fluorescence microscopy showing mitochondrial network in cells growing on graphene and glass substrate. Staining cells with JC-1 demonstrated the influence of hydrogen peroxide on mitochondria.

## Data Availability

The data presented in this study are available within the article.

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
