# Peer review of "Cytocompatibility of Graphene Monolayer and Its Impact on Focal Cell Adhesion, Mitochondrial Morphology and Activity in BALB/3T3 Fibroblasts"

_materials, 2021, doi:10.3390/ma14030643_

Round 1

Reviewer 1 Report

The authors investigated the effect of graphene scaffold on morphology, viability, cytoskeleton, focal contacts, mitochondrial network morphology and activity in BALB/3T3 fibroblasts. This article shows interesting results, but more aspect needs to be improved before I recommend its publication in Materials.

  • Please write the source of graphene used in this work.
  • I suggest to move the Raman results and the microscopy evaluation of materials at the Results section as a separate sub-section Characterization.
  • I suggest to improve the contrast of the Figure 2, and decrease the noise of the background.
  • Please increase the size of Figure 1.
  • How many times did you repeat the tests? Please include this information in the figures caption.
  • Give more details in the legend regarding the images shown in all Figures.

Author Response

First of all, we would like to express our gratitude for valuable remarks.

Graphene used in our experiment has been obtained from Graphenea company, Spain. We put appropriate information into the main text of the article.

Secondly, we considered moving the Raman results and microscopic evaluation in accordance with the second suggestion, but ultimately decided against it, as they only serve as further decription of the material studied and would be out of place in the results section.

Figures and their captions have been updated following remarks.

All tests were done in triplicate. Sufficient information about that has been introduced into the text of the article

Reviewer 2 Report

This work investigated the cytotoxic effect of graphene monolayers to BALB/3T3 cell culture. The morphology of fibroblasts growing and mitochondrial activity of BALB/3T3 cell on graphene scaffold were examined and showed positive results. The manuscript is generally well organized. I would recommend publication with some minor suggestions.  

Comment 1: It has been reported that monolayer graphene could also exhibit antibacterial activities when their edges are exposed to the cell. Did the authors observe any pattern that would suggest the cell growth was inhibited around those edges?

Comment 2: In Figure 5 C and D, how many replicates were conducted? The fact that the error bars are comparable to or even larger than the difference between the control and graphene scaffold raised concerns of whether the differences are statistically significant.  

Comment 3: The increase of mitochondrial activity was attributed to the nano-topographic properties of graphene. I would recommend some further investigation (e.g. AFM) to justify this claim because a continuous layer of graphene would be much smoother than the original glass slide.

Author Response

We appreciate effort put into review of our article. Therefore, we would like to express our gratitude for that. 

In response to the Reviewer's comments:

Graphene in our experiment was applied to the entire surface of a round coverslip. This form of graphene, unlike graphene flakes with many sharp edges, does not damage the cell. We analyzed a number of images and did not observe any damage due to the presence of graphene. In a pilot study (not published), we also used samples of graphene applied as a square to a round coverslip to see if cell-damaging activity could occur at the interface between the two materials. We did not record any damage. In addition, we also performed a comet assay in this experiment and this assay also showed no damage.

Three independent experiments were done in triplicate. For focal contact number per cell at least 50 cells of each experiment were counted, and for focal contacts size at least 10 areas of focal contacts for cell were analyzed. Cells were randomly selected to evaluate the mentioned above parameters: as we can see on Fig. 4 and Fig. 5 A and B cells varied in size and this could be possible explanation of the large error bars.

The mono-layer graphene used in our experiment had wrinkles, which are the main contributor to the graphene roughness. Other authors confirm that: „Nanowrinkled hydrophobic graphene, thus, exhibits superior characteristics for those biomedical applications …” (Verdanowa et al., 2016). Moreover these autors demonstrated: „the greater importance of nanoscale topography than surface wettability” in cell physiology. Images of glass and graphene coating glass using AFM method are available in the article listed below.

Verdanova M, Rezek B, Broz A, Ukraintsev E, Babchenko O, Artemenko A, Izak T, Kromka A, Kalbac M, Hubalek Kalbacova M. Nanocarbon Allotropes-Graphene and Nanocrystalline Diamond-Promote Cell Proliferation. Small. 2016 May;12(18):2499-509. doi: 10.1002/smll.201503749. Epub 2016 Mar 22. PMID: 27000766.

Reviewer 3 Report

The study of the processes of interaction of cells with various surfaces is an important area of research. Progress in this area largely determines the prospects for the application of tissue engineering methods in clinical practice. Therefore, the results presented in the article "Cytocompatibility of graphene monolayer and its impact on focal cell adhesion, mitochondrial morphology and activity in BALB/3T3 fibroblasts" are new and relevant. The presented scientific work uses modern research methods, and the article is written at a high scientific level. However, I have two minor comments to make to the authors of the paper.

  1. It is known that the ability of cells to interact with the substrate by the chemical composition of the surface and its ability to be wetted with water. In this regard, I consider it necessary to supplement the article with a study of these surface parameters in a comparative aspect. I am sure that these studies will significantly improve this work.
  2. It is desirable to supplement the work with a comparative study of surfaces using the SEM or AFM method to understand the differences in the structure of the studied surfaces.

Author Response

Firstly, we'd like to thank the reviewer for taking time to thoroughly read and analyze our work. However, we cannot currently supplement our article the way the reviewer suggests. We have produced another work exclusively devoted to comparing hydrophilicity/hydrophobicity and roughness of two surfaces: graphene on glass and glass by itself. It is now being processed by another journal and we cannot use its material until the article gets published or rejected.

At this moment we can only say that the mono-layer graphene used in our experiment has wrinkles, which are the main contributor to the graphene roughness. Other authors confirm that: „Nanowrinkled hydrophobic graphene, thus, exhibits superior characteristics for those biomedical applications …” (Verdanowa et al., 2016). Moreover, these autors already demonstrated: „the greater importance of nanoscale topography than surface wettability” in cel physiology.

This manuscript is a resubmission of an earlier submission. The following is a list of the peer review reports and author responses from that submission.

Round 1

Reviewer 1 Report

The paper is very clear and welldone  so that it should interesting to go on by further studies in vivo also.

Reviewer 2 Report

The manuscript “Cytocompatibility of graphene monolayer and its impact on focal cell adhesion, mitochondrial morphology and activity in BALB/3T3 fibroblasts” by Lasock et al., presents effects of graphene monolayer on BALB/3T3 cells viability, morphology, focal contact and mitochondrial network. All performed tests suggest no cytotoxicity of graphene layer.

The papers is well presented and clear. The literature data are adequate. However, the paper lacks the novelty as the same team has already published a similar work on another fibroblast cell line L929

Losocka et al., Toxicol in vitro, 2018.

In Discussion the authors declares that the aim was to perform the same study to test whether the effect of cytotoxicity was cell dependent (line 314).

This idea could provide new data in case another cell type were chosen. The second study on fibroblast that provides similar results as the first one (Graphene is not cytotoxic in vitro) lacks novelty.

Thus, I cannot recommend this paper to be published in Nanomaterials.

Reviewer 3 Report

Review of Cytocompatibility of Graphene Monolayer and Its Impact on Focal Cell Adhesion, Mitochondrial Morphology and Activity in BALB/3T3 Fibroblasts

Comments to the authors

This is a well written manuscript with nice supportive data and I could find little if anything to fault it. My comments below are very trivial.

I am not sure what Fig 2 is supposed to show. In the manuscript sent to me for review I have a blank blue coloured image and from the legends I am unsure what this is supposed to represent or whether the figure did not transfer over correctly. The authors need to clarify this.  If this figure is meant to convey the uniform transfer of grapheme on to slides surely this could be simply conveyed by a simple sentence.

Line 170 define PFA

Line 171 is this Triton-X-100 ?